

# EUSO-SPB2: A sub-orbital cosmic ray and neutrino multi-messenger pathfinder observatory

Austin Cummings[1], Johannes Eser[2*], George Filippatos[3], Angela V. Olinto[2],
Tonia M. Venters[4] and Lawrence Wiencke[3] for the JEM-EUSO collaboration

**1** Department of Astronomy and Astrophysics, Pennsylvania State University,
University Park, PA, USA
**2** Department of Astronomy and Astrophysics, The University of Chicago, Chicago, IL, USA
**3** Department of Physics, Colorado School of Mines, Golden, CO, USA
**4** Astroparticle Physics Laboratory, NASA Goddard Space Flight Center, Greenbelt, MD, USA

⋆ jeser@uchicago.edu

## Abstract

The next generation of ultra-high energy cosmic ray (UHECR) and very-high energy neutrino observatories will address the challenge of the extremely low fluxes of these particles at the highest energies. EUSO-SPB2 (Extreme Universe Space Observatory on a Super Pressure Balloon 2) is designed to prepare space missions to address this challenge. EUSO-SPB2 is equipped with 2 telescopes: the Fluorescence Telescope, which will point downwards and measure fluorescence emission from UHECR air showers with an energy above 2 EeV, and the Cherenkov Telescope (CT), which will point towards the Earth's limb and measure direct Cherenkov emission from cosmic rays with energies above 1 PeV, verifying the technique. Pointed below the limb, the CT will search for Cherenkov emission produced by neutrino-sourced tau-lepton decays above 10 PeV energies and study backgrounds for such events. The EUSO-SPB2 mission will provide pioneering observations and technical milestones on the path towards a space-based multi-messenger observatory.

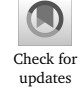

## 1 Introduction

Future space-based missions such as the proposed Probe for Extreme Multi-Messenger Astophysics (POEMMA) [1], aim to overcome the shortcomings of ground-based observation by increasing the detection volume by several orders of magnitude by looking down on the Earth's

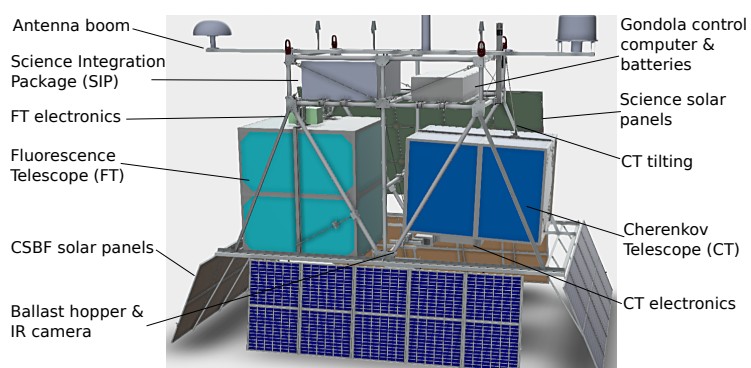

Figure 1: Design drawing of the EUSO-SPB2 payload, consisting of two Telescopes (Fluorescence and Cherenkov), an IR camera and components needed for a long duration stratospheric balloon flight.

atmosphere, making these detectors the next promising step in Ultra High Energy Cosmic Ray (UHECR, E > $10^{18}$ eV=1 EeV) and Very High Energy Neutrino (VHE, E > $10^{15}$ eV=1 PeV) astronomy.

These cosmic particles are of interest as UHECRs are the most energetic particles known to exist with measured energies exceeding 100 EeV but with an extremely suppressed flux (see [2] and references therein) leaving their sources and acceleration mechanisms still unknown after decades of ground observations. VHE neutrinos, on the other hand, are a second messenger to help understand astrophysical sources and other energetic processes in the universe. As chargeless, weakly interacting particles, they point back to their sources however their almost vanishing interaction cross section require an enormous detector volume. Only a few cosmic neutrinos with energies above 1 PeV have been detected at the time of this writing.

The Extreme Universe Space Observatory on a Super Pressure Balloon 2 (EUSO-SPB2) is a pathfinder for future space-based detectors, which will raise the technological readiness and verify the proposed detection techniques in near-space environments without the risk and cost of an actual space mission.

Besides the R&D work, EUSO-SPB2 has the following main scientific objectives:

1. Observe the first extensive air showers (EASs) via the fluorescence technique from suborbital space.
2. Observe Cherenkov light from upward-going EASs initiated by cosmic rays.
3. Measure the background conditions for the detection of neutrino induced upward-going air showers.
4. Search for neutrinos from astrophysical transient events (e.g. binary neutron star mergers).

## 2 The mission and instrument

EUSO-SPB2 is designed as a payload for a NASA super pressure balloon with a targeted flight duration of up to 100 days at a nominal float altitude of 33 km (Fig. 1). The targeted launch date is Spring 2023 from Wanaka, NZ, allowing the balloon to circulate around the globe following a stratospheric air current[1] developing twice a year at latitudes of ∼ 45°S. Five hours of darkness can be expected on average every night during flight which is the time EUSO-SPB2 can take data.

---

[1]https://earth.nullschool.net/#2017/04/30/2300Z/wind/isobaric/10hPa/orthographic=-177.79,-57.63, 408/loc=171.065,-43.450

To satisfy not only the R&D requirements of such a mission, but also the predefined science goals, EUSO-SPB2 has 2 different telescopes, each optimized for its respective task:

1. **The Fluorescence Telescope (FT)** has a modular 6912 pixel camera based on Multi-anode Photo Multiplier Tubes (MAPMTs: R11265-M64-203 from Hamamatsu) with a $1\mu s$ integration time, capable of single photoelectron counting with a double pulse resolution of 6 ns. BG3 filters[2] narrow its sensitivity to the UV region (290-430 nm) and reduce the background of photon counts outside this optimal frequency band. The optical system is a Schmidt design with 6 identical spherical mirror segments and a 1 m diameter aperture yielding a 12° by 36° field of view fixed to the nadir.

2. **The Cherenkov Telescope (CT)** has a 512 Silicon PhotoMultiplier (SiPM: S14521-6050AN-04 from Hamamatsu) pixel camera design to record very fast and bright signals, like the Cherenkov emission from air showers. The integration time of this camera is 10 ns, and it will be the first time for such an instrument to be flown on a stratospheric balloon. The optical system is very similar to the FT but only contains 4 mirror segments which are aligned in such a way that the light is focused in two distinct spots on the camera, helping to distinguish between direct cosmic ray hits (only one spot) and light from outside the telescope (2 spots), thereby reducing the background. The field of view is 6.4° in zenith and 12.8° in azimuth and can be pointed during flight from horizontal all the way to 10° below the Earth's limb depending on the targeted particle detection and its science goals. The goal for above-the-limb observation is the measurement of cosmic rays above 1 PeV, where the event rate is expected to be large (hundreds of events per hour of live time). The goal for the below-the-limb observation is to search for signals corresponding to the upward $\tau$-induced EAS and measure backgrounds for such events, as well as to follow up on alerts of astrophysical events occurring during the mission, such as neutron star-neutron star mergers sec. 3.3).

The payload also contains an IR camera to monitor for clouds in the field of view of the FT since they have significant impact on the exposure and reconstruction of recorded EAS candidates. A more in depth description of the individual telescopes is given in [3] and [4].

## 3 Expected performance

The expected performance of EUSO-SPB2 is estimated by extensive Monte Carlo simulation. While for the FT, the detector response is simulated in detail using GEANT4, the detector response for the CT is estimated based on best estimates of the single component performance. In the near future, the simulation will be updated for both instruments using measured detector performance as evaluated during ground testing.

### 3.1 UHECR via fluorescence technique

As a pathfinder for future space detection of UHECR, the primary goal for the EUSO-SPB2 FT is the novel detection and reconstruction of a cosmic ray via the fluorescence technique from suborbital space, verifying the feasibility of the technique. The expected performance of the FT is evaluated by an extensive simulation study using 1.6 million showers (only proton primary) landing on a disk with 100 km radius centered on the detector location projected on ground. The zenith angle distribution follows $\sin\theta\cos\theta$ (0°- 80°) and the simulated energy range is between $10^{17.8}$ eV and $10^{19.7}$ eV. To simulate a realistic flight, background measurements from

---

[2]https://www.schott.com/shop/advanced-optics/en/Matt-Filter-Plates/BG3/c/glass-BG3

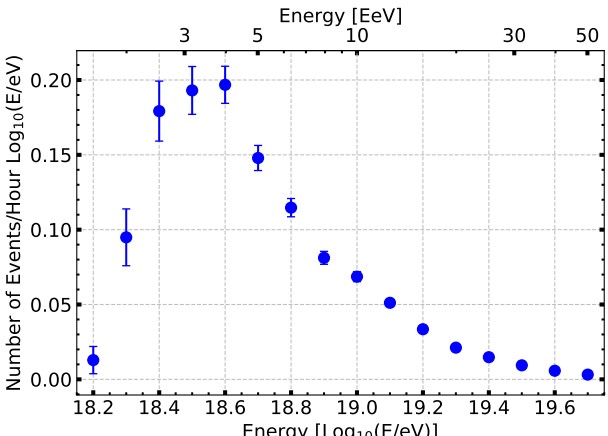

Figure 2: Expected UHECR event rate for the EUSO-SPB2 FT. The energy threshold is estimated to be $\sim 2\,\mathrm{EeV}$ with peak sensitivity at $4\,\mathrm{EeV}$ [5]. The Auger UHECR energy spectrum [6] is used to convert a trigger rate to an hourly event rate.

previous experiments are scaled appropriately to the new detector characteristics leading to an average of 1count/px/GTU. It has to be noted that a clear sky is assumed, no cloud impact is considered.

We expect 0.12 events per observation hour by integrating the entire energy span as shown in Fig. 2 with a pure statistical error of 0.01 events. Due to limited telemetry, the identified event candidates will be prioritized using machine learning techniques. Details of the trigger and the prioritization can be found in [5,7].

Additionally, EUSO-SPB2 will search for the candidate upward-going events similar to the ones reported by the Antarctic Impulse Transient Antenna (ANITA) [8] but with even steeper emergence angles. A detailed study of the sensitivity of EUSO-SPB2 to such events is ongoing.

## 3.2 Direct cosmic rays via Cherenkov technique

To measure Cherenkov emission from EAS from suborbital space, EUSO-SPB2 will point its CT above the limb. In this configuration, EUSO-SPB2 will observe cosmic rays developing primarily in rarefied atmosphere. In such an environment, the atmospheric attenuation is minimized, but the threshold for Cherenkov emission is higher, leading to an interesting behavior where the maximum sensitivity to above-the-limb cosmic rays is energy dependent and occurs distinctly above the horizon (with higher energy cosmic rays able to compete with atmospheric extinction and be observed closer to the horizon). The expected energy threshold for such events is around 1 PeV, primarily due to the highly forward-beamed nature of EAS Cherenkov emission (noting particularly that the emission angle in rarefied atmosphere grows exceedingly small, allowing for even more intense beaming). The nature of the cosmic ray flux ensures that the majority of these events are expected to be observed with energies near the detection threshold, and therefore, a few degrees above the limb. Atmospheric refraction of emission at the limb is expected to be $\sim 1°$, so measurement of these events will help to clarify how strong this refraction is. A detailed discussion of the simulation used is given in [9].

The cumulative event rate for above-the-limb cosmic rays is presented in Fig. 3, and shows more than 100 events per hour of data taking with an energy threshold as low as 1 PeV. This guaranteed and frequent signal will allow for validation of the technique and will help to refine detection methods and build reconstruction tools for possible neutrino observation.



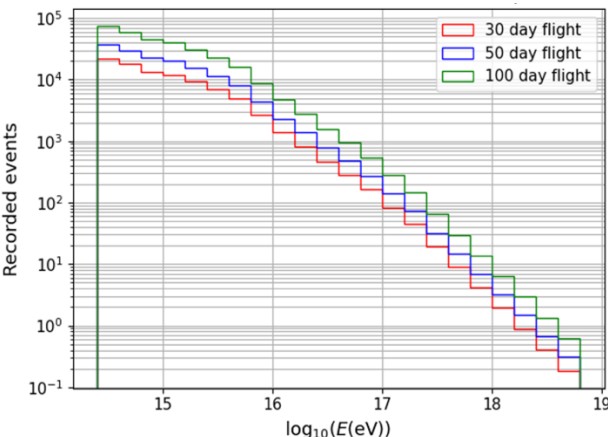

Figure 3: Expected, cumulative event rate for above-the-limb cosmic rays [9]. 10 photons m$^{-2}$ns$^{-1}$ is the detection threshold applied in this calculation.

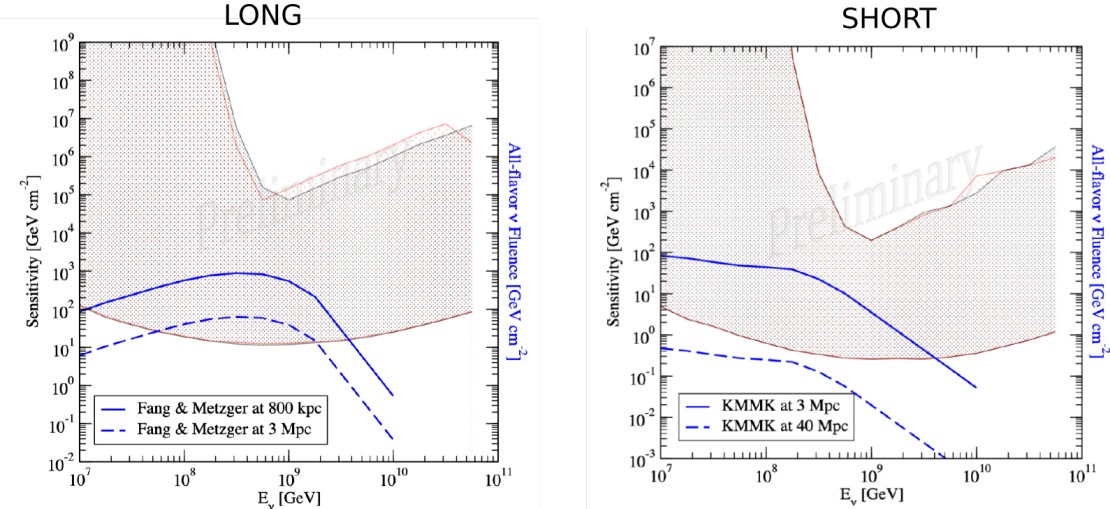

Figure 4: Sensitivity to ToO neutrino fluence for the EUSO-SPB2 Cherenkov telescope [11]. Two source models are considered: long-burst [12] and short-burst [13]. In case of the long-burst scenario, a flight duration of 30 days (red line) and 100 days (black line) are considered.

## 3.3   Neutrinos from target of opportunities

One unique capability of EUSO-SPB2 is that the CT can follow up on a large variety of astrophysical event alerts, countering the fact that the sensitivity of EUSO-SPB2 to the diffuse neutrino flux is lower than that of current ground based experiments, primarily due to the small field of view and the short mission duration [10]. The sensitivity for 2 different source scenarios is shown in Fig. 4) The long-burst model, which lasts days to weeks for which the neutrino fluence is averaged over that time and 2) the short-burst model, which lasts around 1000 s and it is assumed that the source is in the FoV for the entire time. As the effect of the Sun and the Moon are negligible for the short-burst scenario, they are considered only for the long-burst models. For both source models, the movement of the balloon is not considered, but will be included in future simulations.  For the long-burst scenario, the emission model from [12] for Binary Neutron star mergers was chosen scaled for distances of 0.8 Mpc and 3 Mpc (see left panel). The right panel shows the sensitivity for moderate emission of short gamma ray bursts per the model of [13]. The distances 3 Mpc and 40 Mpc are chosen as Grav-

itational Waves have been received from these distances. It must be noted that the relatively short mission duration of EUSO-SPB2 reduces the probability of such close proximity events occurring.

## 4 Conclusion

A successful launch and flight of EUSO-SPB2 will raise the technological readiness for future space missions while also accomplishing measurements for the first time from suborbital space. The planned launch is scheduled from Wanaka, NZ in early 2023.

Extensive simulation studies have been conducted to evaluate the performance of both telescopes on board EUSO-SPB2. These simulations have demonstrated that the FT will record 0.12 UHECR events per hour while also looking for ANITA event-like candidates. While the CT does not have a sensitivity to the diffuse neutrino flux (cosmogenic) that is competitive with existing ground based experiments, it has the capability to follow up on astrophysical events, increasing the chance of neutrino observation. In addition, the CT will measure hundreds of direct cosmic rays while pointing above the limb. These events will not only validate the performance of the camera during flight but due to being very similar to neutrino induced showers, will allow the development and enhancement of trigger, detection and reconstruction techniques for future missions.

## Acknowledgments

**Funding information** The authors would like to acknowledge the support by NASA award 11-APRA-0058, 16-APROBES16-0023, 17-APRA17-0066, NNX17AJ82G, NNX13AH54G, 80NSSC18K0246, 80NSSC18K0473, 80NSSC19K0626, 80NSSC18K0464, 80NSSC22K1488, 80NSSC19K0627 and 80NSSC22K0426.

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
