# Peer review of "EUSO-SPB2: A sub-orbital cosmic ray and neutrino multi-messenger pathfinder observatory"

_SciPost Physics Proceedings, doi:SciPost Phys. Proc. 13, 038 (2023)_

## Round 1 · Referee Report · Anonymous · 2023-1-12

Strengths
See "report"
Weaknesses
See "report"
Report
This proceeding provides a comprehensive overview of the instrumentation, the expected signal signatures and their rates, and the science case of the EUSO-SPB2 mission.
The description of the future instrumentation of the SPB2 gondola with EUSO R&D components is convincing and the explanations of the design decisions are reasonable to follow. The use of new types of SiPMs (even in the stratosphere in this case!) could be highlighted a bit more, as these sensors are increasingly important in low-light-level sensing in many different fields of research.
The Subchapter discussing the expected event rates and thus detectable events is convincing - not least because the limitations of this EUSO pathfinder are pointed out right away.
The introduction of a second detection channel, (astrophysical) neutrinos along with UHECRs, is compelling and expands the science case of EUSO-SPB2 substantially. Also the potential possibility to detect further up-going events like the one seen by ANITA is interesting and makes curious.
The mention of the capability of EUSO-SPB2 to potentially respond to alerts from other experiments and observatories during the flight period is reasonable since it expands even more the possible goals and outcomes of this mission.
The content of this proceedings is well thought out, interesting for many readers and describes well the status and science goals (and R&D goals) of EUSO-SPB2 within the small number of pages of a proceeding. The signal rates and sensitivities simulated via GEANT4 and MC are within a realistic range, presumably also incorporating experiences and findings from the previous EUSO-SPB1 flight.
During the review process, text errors and their suggested corrections, as well as possible small improvements to the Figures, were provided to the authors via anonymized file (uploaded) here.
Author: Johannes Eser on 2023-01-18 [id 3247]
(in reply to Report 1 on 2023-01-12)First we would like to thank the reviewer for the careful read and the useful suggestion which were incorporated in the new version.
The only unaddressed comment is the suggestion to incorporate previous flight path in section 2. We decided to add a link to a website showing the stratospheric air current mentioned as a footnote instead. Hopefully that is satisfactory as there is no way to add another figure within the page limit even trying the suggestion of aligninmg fig2 and 3 next to each other.

---

## Editorial Decision

published